# Bovine Papillomavirus Type 1 or 2 Virion-Infected Primary Fibroblasts Constitute a Near-Natural Equine Sarcoid Model

**DOI:** 10.3390/v14122658

**Published:** 2022-11-28

**Authors:** Edmund K. Hainisch, Christoph Jindra, Paul Reicher, Lea Miglinci, Daniela M. Brodesser, Sabine Brandt

**Affiliations:** 1Research Group Oncology, Equine Surgery, Equine Clinic, University of Veterinary Medicine, 1210 Vienna, Austria; 2Division Molecular Oncology and Haematology, Karl Landsteiner University of Health Sciences, 3500 Krems an der Donau, Austria

**Keywords:** horse, sarcoid, bovine papillomavirus, BPV1, BPV2, virions, primary fibroblasts, in vitro model

## Abstract

Equine sarcoids are common, locally aggressive skin tumors induced by bovine papillomavirus types 1, 2, and possibly 13 (BPV1, BPV2, BPV13). Current in vitro models do not mimic de novo infection. We established primary fibroblasts from horse skin and succeeded in infecting these cells with native BPV1 and BPV2 virions. Subsequent cell characterization was carried out by cell culture, immunological, and molecular biological techniques. Infection of fibroblasts with serial 10-fold virion dilutions (2 × 10^6^-20 virions) uniformly led to DNA loads settling at around 150 copies/cell after four passages. Infected cells displayed typical features of equine sarcoid cells, including hyperproliferation, and loss of contact inhibition. Neither multiple passaging nor storage negatively affected cell hyperproliferation, viral DNA replication, and gene transcription, suggestive for infection-mediated cell immortalization. Intriguingly, extracellular vesicles released by BPV1-infected fibroblasts contained viral DNA that was most abundant in the fractions enriched for apoptotic bodies and exosomes. This viral DNA is likely taken up by non-infected fibroblasts. We conclude that equine primary fibroblasts stably infected with BPV1 and BPV2 virions constitute a valuable near-natural model for the study of yet unexplored mechanisms underlying the pathobiology of BPV1/2-induced sarcoids.

## 1. Introduction

Papillomaviruses (PVs) are relatively small non-enveloped DNA viruses that can induce benign or malignant tumors in humans and animals. PVs consist of an icosahedral capsid harboring a circular, double-stranded DNA genome. The capsid is composed of 72 major L1 capsid protein pentamers termed capsomers, and a minimum of 12 minor L2 capsid protein monomers. The viral genome contains open reading frames (ORF) for early (E) regulatory proteins (notably E1, E2 and E4) and two to three oncoproteins (E6, E7 and frequently E5), and for the late (L) capsid proteins L1 and L2. A non-coding region termed long control region (LCR) is located between the L1 and E6 ORFs and provides elements required for viral replication and transcription [1]. Early research in natural animal models has revealed that PVs are usually host-specific and have a pronounced tropism for cutaneous and mucosal keratinocytes [2]. The genus delta-(δ-) PVs substantially differs from other PV genera. It regroups a small number of PVs that also or mainly infect dermal fibroblasts and-perhaps for this reason-have a wider host range. This feature is best documented for bovine δ-PV type 1 (BPV1), which, alike BPV2 and 13, also infects horses and other equids [3,4,5]. BPV1, 2 and 13 are closely related virus types as shown by their high genetic similarity reaching >90% on L1 capsid protein level [5].

In bovids, BPV1, 2 and 13 infect the epidermis and underlying dermis and give rise to benign fibropapillomas (“cow warts”) that usually regress spontaneously after several months. Infection is productive in the epidermal part of the lesions, with terminally differentiated keratinocytes containing high quantities of infectious virions. In the dermal part, fibroblasts harbor viral DNA in an episomal form and do not support virion formation [6].

In horses and other equids such as donkeys, mules, or zebras, BPV types 1, 2, and probably 13 are the causative agents of benign, yet locally aggressive skin tumors termed sarcoids [3,4]. With a worldwide prevalence of up to 12%, sarcoids constitute the most common neoplastic disease in horses [7,8]. Sarcoids are classified into six clinical types, ranging from mild-type occult, verrucous, or nodular lesions to ulcerated fibroblastic, mixed, or malevolent sarcoids [9]. Despite various attempts to identify factors that determine the clinical presentation of sarcoids, this issue remains unresolved [10,11].

In sarcoids, BPV1/2 infection predominantly involves dermal fibroblasts and is episomal in these cells to assure genome maintenance. Importantly, viral episomes replicate in synchrony with cell division [12,13]. This efficient form of PV amplification and the ability of BPV1 and BPV2 to establish latent episomal infection also in apparently healthy skin of sarcoid-affected horses [14,15,16,17,18] likely explain the frequently reported onset, progression or reoccurrence of sarcoids following accidental or iatrogenic trauma [3,19,20,21]. The role of trauma in (re-) activation of PV-induced disease is widely accepted today [22]. However, it remains completely unclear, how viral infection can spread within the horse’s body and eventually involve the entire integument.

Given the etiological association of various PV types with human and animal cancers, considerable efforts are put into modelling PV infections and related diseases since many years. These efforts were long time hampered by the usual species-specificity of PVs, and the dependence of their productive life cycle on the differentiation program of keratinocytes [23]. Today, there are a series of in vitro and in vivo models able to recapitulate important aspects of PV infection and related carcinogenesis, e.g., organotypic raft cultures, xenograft systems, PV transfected cells, natural animal PV models and transgenic mice [23,24]. In addition, PV L1 (L2) virus-like particles (VLPs), pseudovirions (PsV), and quasivirions (QV) have been developed to study human PV (HPV) infection and transmission pathways [25]. Despite these considerable advances, current models continue to have various limitations that have to be born in mind when reaching conclusions as to how PV infection and related disease are regulated in vivo [23]. Notably recombinant particles were meanwhile shown to significantly differ from native PV virions with respect to the biology of PV infection [25].

BPV1 is the best-studied animal PV and BPV1 research has greatly helped clarify key functions of PV early proteins [6]. Yet, the mechanisms underlying de novo infection of equids by BPV1 (or other δ-BPVs) and propagation of the virus within the equid host are still unknown. So far, available sarcoid in vitro models are based on primary sarcoid cells and equine fibroblasts transfected with parts of or the entire BPV1 genome. These models significantly helped advance our understanding of BPV1-mediated neoplastic transformation [26,27,28,29,30,31]. Yet, they do not maintain episomal infection upon storage and are unsuited for the study of infection pathways. All attempts to experimentally infect cultured equine fibroblasts with wild-type BPV1 virions were unsuccessful so far [29,32,33].

Herein we describe the successful infection of equine primary fibroblasts with wild-type BPV1 and BPV2 virions, and the study of these cells with respect to growth characteristics, stability of infection, viral gene transcription, viral DNA secretion, and release of extracellular vesicles (EVs) harboring viral DNA.

## 2. Materials and Methods

**Generation of equine primary fibroblasts.** Skin was excised immediately post-mortem from the axilla, eye lid, and prepuce of horse “Arc” euthanized for sarcoid-unrelated reasons (colic) with the owners’ written consent. Skin material was immediately placed in medium (Dulbecco’s Modified Eagle Medium with high glucose [4.5 g/L] and GlutaMAX, 1% antibiotic-antimycotic mixture [anti-anti], 50 μg/mL Fungizone and 250 μg/mL Gentamycin; all from Life Tech/Thermo Fisher Scientific, Vienna, Austria) supplemented with 10% heat-inactivated fetal bovine serum [FBS-HI] and transferred to the laboratory. Then, sample material was washed, cut into small pieces, and digested in 2.5% Trypsin solution diluted 1:10 in HBSS (both from Life Tech/Thermo Fisher Scientific) at 4 °C overnight, followed by incubation at 37 °C for 30 min. Digested tissue was filtered and scraped through a 100 μm Cell Strainer (Pluriselect, Eichgraben, Austria), collected in FBS, and rinsed with HBSS. Then, cells were harvested by centrifugation, washed thoroughly, and counted using a Nucleo-Counter NC-250 System (ChemoMetec, Kaiserslautern, Germany) according to instructions of supplier. Then, cells were seeded at a density of 10,000 viable cells/cm^2^ in six-well plates containing 1 mL culture medium (Dulbecco’s Modified Eagle Medium with high glucose [4.5 g/L] and GlutaMAX and 10% heat-inactivated fetal bovine serum [FBS-HI]; all from Life Tech/Thermo Fisher Scientific, Vienna, Austria) at 37 °C with 5% CO_2_. Primary fibroblasts received the designations ArcA, ArcL, and ArcP according to their donor and the respective sampling sites. Fibroblasts were expanded via splitting of confluent cells and passaging. For short- and long-term storage, cells were harvested, washed, pelleted, resuspended in freezing medium (culture medium supplemented with 30% FBS and 10% DMSO [Sigma-Aldrich, Vienna, Austria]) and immediately transferred to −80 °C. After 12 h, cells were stored at −150 °C. To take frozen cells back into culture, they were thawed in a water-bath at 37 °C, transferred to a 15 mL centrifuge tube supplied with 11 mL pre-warmed culture medium, pelleted at 1.000 rpm for 5 min at room temperature and then resuspended in 1 mL pre-warmed culture medium containing 10% FBS. Then, cells were seeded in T75 flask containing 20 mL culture medium and incubated at 37 °C with 5% CO_2_.

**Virion preparation and quantification.** Cow warts were obtained by therapeutic excision from affected heifers in Austria (BPV1) and Portugal (BPV2). Presence of BPV1 or BPV2 DNA was ascertained by PCR [34] from purified cow wart DNA (DNeasy Blood and Tissue Kit; Qiagen, Hilden, Germany) and subsequent amplicon sequence analysis (Eurofins, Vienna, Austria). Wild-type BPV1 and BPV2 virions were purified from cow warts according to an established protocol [35]. Concentrations of infectious virion were determined by in vitro focus formation assay [36,37].

**Infection of fibroblasts with virions**. ArcA, ArcL and ArcP cells were seeded in six-well plates (150,000 cells per well) and incubated at 37 °C with 5% CO_2_ overnight. On the next day, wells were rinsed twice with phosphate-buffered saline pH 7.2 (PBS) and supplied with 2 × 10^6^ to 20 BPV1 or BPV2 virions in FBS-free medium and incubated for two hours. Native Arc fibroblasts served as non-infected control cells. Then, cells were washed, supplied with medium containing 10% FBS and expanded as described above. Cell aliquots from every passage were stored as native pellets or in TRIzol (Invitrogen/Thermo Fisher Scientific) at −20 °C. Infected Arc and control cells were cultured for 23 to 34 passages. They were closely monitored with respect to growth characteristics and morphological changes by phase-contrast microscopy. Relative proliferation of infected versus non-infected fibroblasts throughout passages was assessed by determining respective numbers of viable cells using the NucleoCounter NC-250 System (ChemoMetec, Allerod, Denmark) according to instructions of supplier.

**Viral DNA detection.** Infected fibroblasts and non-infected control cells were harvested and subjected to DNA extraction using a DNeasy Blood and Tissue Kit (Qiagen). Following successful equine beta-actin (EBA) PCR [38] confirming the PCR-compatible quality of all DNA extracts were subjected to consensus BPV1/2 E5 and L1 PCR as described previously [38]. Amplification products (16 µL) were analysed by gel electrophoreses on 1.5% Tris-acetate agarose (TAE) gels and visualized by ethidium bromide staining. Viral DNA loads per BPV1-infected cell were determined by BPV1 E2 qPCR according to an established protocol [11]. BPV1-positive sarcoid DNA, non-infected Arc DNA, and sterile water served as respective positive, negative, and no-template (ntc) controls in all PCR and qPCR experiments.

**Screening for virions by immunocapture PCR (IC/PCR).** BPV1 and BPV2-infected fibroblasts, non-infected control cells, as well as respective culture supernatants were harvested. Cells were pelleted and resuspended in 500 µL PBS/pellet. Then, cells were minced mechanically and centrifuged at 13.000 rpm for 5 min at 4 °C. Obtained cell extract supernatants were transferred to new 1.5 mL tubes and stored at −20 °C overnight. MμlTI Ultra PCR tubes (0.65 mL; Sorenson™ BioScience, Inc., Salt Lake City, UT, USA) were coated with 60 μL of L1 monomer-specific antibody 1H8 (Abcam, Cambridge, UK) or L1-capsomer-specific antibody 5B6 (kindly provided by R. Roden, The Johns Hopkins University, Baltimore, MD, USA) diluted 1:50 in 0.1 M sodium carbonate buffer pH 9.6 and incubated at 4 °C overnight. Subsequently, tubes were washed twice with PBS-Tween (0.1% Tween 20) and rinsed thrice with PBS. Then, 50 μL aliquots of culture supernatants, cell extract supernatants, sarcoid extract supernatant (positive control; +c) or sterile water (ntc) were applied to the pre-coated tubes and incubated at 4 °C overnight. After washing thoroughly as described above, consensus BPV1/2 E5 PCR was carried out [15]. Sixteen μL aliquots of resulting amplification products were visualized on 1.5% TAE gels by ethidium bromide staining.

**Detection of viral transcripts**. BPV1-infected cells from different passages (P11, P12, P23) were harvested, pelleted, and RNA extraction was carried out according to the TRIzol RNA isolation protocol provided by the manufacturer (Invitrogen).). DNase 1 digestion and enzyme inactivation were performed in RNeasy microcolums (Qiagen), as recommended. DNA-free RNA isolates were eluted in 22 μL of RNase-free water and cDNA synthesis was conducted in MμlTI Ultra PCR tubes (Sorenson Bioscience), each containing 50 mM Tris-HCl, pH 8.3, 75 mM KCl, 3 mM MgCl2, 10 mM DTT, 2 mM of each dNTP, 200 pmol random hexamer (Roche Diagnostics, Vienna, Austria), 20 U of recombinant RNase inhibitor (RNaseOUT; Invitrogen/Thermo Fisher Scientific), 9.5 μL of RNA and 200 U of SuperScript III reverse transcriptase (+RT; Invitrogen/Thermo Fisher Scientific, Vienna, Austria) or 1 μL of 87% glycerol as mock RT control (−RT) in a total volume of 20 μL. After incubation for 1 h at 50 °C in a thermocycler (LifeTouch Thermal Cycler, Biozym, Hessisch-Oldendorf, Germany; also used for all standard PCR experiments) 2-μL aliquots of each reaction mixture were subjected to E5 and L1 PCR and subsequent gel electrophoresis according to Brandt et al. [15,38].

**Screening for transmission of infection.** To analyze whether BPV1 DNA would be transmissible via cell secretions, supernatant was collected from confirmedly BPV1-infected ArcA cells (ArcAi-1; P10) filtered through 0.22-µm Millex^®^-GP filters (Millipore Cooperation, Billerica, MA, USA) to remove cells and cell debris and then used for incubation with non-infected ArcA. After three passages, ArcA were harvested, and subjected to DNA extraction using a DNeasy blood and Tissue Kit. In analogy, DNA was isolated from ArcAi-1 and cell-free supernatant thereof. Then, 2-µL DNA aliquots were PCR-screened for the presence of a 1701 bp E2E5 region of the BPV1 genome using primers 5′E2-2585 (5′-ttgacgaggaggaggatagtgaagagg-3′) and 3′E5-4285 (5′- tacctttggtatca catctggtgg), with numbers corresponding to the position of the first nucleotide of each primer according to Genbank Sequence X02346. To screen supernatant-exposed ArcA for the presence of the entire BPV1 genome (7945 bp), inverse PCR (INV-PCR) was carried out from 2 µL of 1:1 and 1:10 diluted DNA aliquots using back-to-back primers 5´BPV1-6046 and 3´BPV1-6045 [15]. Both PCR reactions were carried out with Phusion Hot Start DNA polymerase according to instructions of manufacturer (Thermo Fisher Scientific). The cycling program for amplification of the 1701 product consisted of 98 °C for 2 min, followed by 45 cycles of 98 °C for 15 s, 68 °C for 30 s and 72 °C for 1 min and a final elongation step at 72 °C for 5 min. For INV-PCR, the elongation time was adjusted to 4 min. Sixteen μL aliquots of resulting amplification products were visualized on 1.5% TAE gels by ethidium bromide staining.

**Purification and screening of EVs for BPV1 DNA.** We finally addressed the possibility that supernatant-mediated transfer of viral DNA from BPV1-infected donor to native recipient fibroblasts would be carried out by EVs. We hence purified EV fractions enriched for apoptotic bodies (ABs), microvesicles (MVs) and exosomes (EXOs) from filtered (0.22-µm Millex^®^-GP filters; Millipore Cooperation), cell-free supernatant of two different BPV1-infected ArcAi-1 cultures in passages 22 and 20 by differential ultracentrifugation as described previously [39]. Subsequently, pelleted EV species were subjected to DNA isolation followed by BPV1 E5 PCR [38](EVs from P22) or amplification the viral oncogenes E6 and E7 (1320 bp; EVs from P20). The latter reaction was achieved using primers 5′LCR-7495 (5′-ctctctgtcggaaccagaactgg-3′) and 3′E7-869 (5′-gctacctttatcgtttgccatgacg-3′) and Phusion Hot Start DNA polymerase (Thermo Fisher Scientific) following instructions of manufacturer. The cycling program consisted of 98 °C for 2 min, followed by 45 cycles of 89 °C for 15 sec, 69 °C for 30 sec and 72 °C for 1 min, and 72 °C for 5 min. BPV1-positive sarcoid DNA, non-infected ArcA DNA and sterile water served as positive, negative, and no-template controls. Sixteen μL aliquots of resulting amplification products were visualized on 1.5% TAE gels by ethidium bromide staining.

## 3. Results

### 3.1. Infection of Equine Primary Fibroblasts with BPV1 and -2 Virions Induced Hyperproliferation, Loss of Contact Inhibition and Most Probably Immortalization

As early as two days post infection, marked hyperproliferation of BPV1- and BPV2-infected equine primary fibroblasts (ArcAi-1, ArcAi-2) compared to non-infected cells was observed. Ten days post infection, numbers of infected cells had increased fivefold in comparison to non-infected cell equivalents. At the same time, infected cells exhibited an increasing loss of contact inhibition (CI) (Figure 1, left), began to pile up or form spherical cell aggregates resembling sarcoid tissue (Figure 1, middle and right). Hyperproliferation, loss of CI and cell aggregation were uniformly noted for BPV1- and BPV2-infected primary equine fibroblasts derived from axilla, eye lid and prepuce throughout culture of these cells until final P34. In contrast, respective non-infected control cells exhibited increased hypoproliferation and apoptosis over passaging. Freezing and re-culturing of infected cells had no negative effect on viral genome maintainance. This contrasts with primary sarcoid cells and BPV1-transfected equine fibroblasts, which do not maintain viral DNA upon storage (S. Brandt; L. Nasir, University of Glasgow, UK; personal communication).

### 3.2. Low Virion Numbers Sufficed to Produce Infection

To address the minimum amount of infectious wild-type BPV1 and BPV2 virions required for successful infection of primary equine fibroblasts, the latter were incubated with decreasing virus concentrations, i.e., 2 *×* 10^6^ to 20 virions. Then, cells were monitored with respect to the presence of viral DNA by E5 PCR over up to 34 passages. As exemplarily depicted, 20 infectious virions (as determined by FFA) sufficed to produce stable infection (Figure 2).

### 3.3. Viral DNA Loads Reached a Physiological Level after Four Passages

At passage 2 (P2), virion concentrations used for infection directly correlated with cell proliferation rates as determined by qPCR, and the higher were cell proliferation rates as determined by comparative automated counting of viable cells. A similar result was obtained for BPV2. After four passages however, average viral DNA loads settled at a level of around 150 copies per cell irrespective of the number of BPV1 or BPV2 virions used for initial infection, as exemplarily shown for ArcAi-1 (Figure 3). This level is likely physiological, as it remained stable throughout all further passages. Uninfected control cells and sterile water (ntc) tested negative for BPV1 and BPV2 DNA, as anticipated.

### 3.4. Viral Genes E5 and L1 Were Transcribed in Infected Fibroblasts

Growth characteristics of infected cells, i.e., hyperproliferation, loss of CI, and assumed immortalization were highly indicative for stable BPV1 and BPV2 DNA replication and expression of transforming proteins. To confirm this assumption, we assessed infected cells from different passages for transcription of the E5 oncogene and the L1 major capsid gene. We expected to detect E5 mRNA throughout passages but speculated that L1 might not be transcribed since episomal infection is thought to be abortive [6]. As exemplarily depicted for ArcAi-1 at P11, E5 mRNA was detected from all infected cell aliquots irrespective of virion concentrations used for initial infection (Figure 4, top panel). Interestingly, this also applied to L1 mRNA (Figure 4, bottom panel). E5 and L1 transcription was seen in BPV1- and BPV2-infected fibroblasts throughout passages (not shown), consistent with stable infection by these virus types.

### 3.5. Infected Fibroblasts Were Not Permissive for Productive BPV1 and BPV2 Infection

Given that L1 expression was consistently detected throughout passaging of BPV1- and BPV2-infected equine fibroblasts, we wished to ascertain that no virions are produced in these cells in line with the concept of an episomal PV infection. To this aim, cell extract and cell culture supernatants were subjected to E5 immunocapture PCR using anti-L1 antibodies 1H8 (monomer-specific) or 5B6 (capsomer-specific). Except for positive controls, i.e., virion suspension (IC_+c_) and sarcoid DNA (+c), all samples scored negative, as exemplarily shown for IC/PCR conducted with capture antibody 5B6 (Figure 5).

### 3.6. BPV1 DNA Could Be Transferred from Infected to Non-Infected Cells via Cell-Free Culture Medium

As stated in the introduction, BPV1 infection can spread within the horse’s body and eventually involve the entire integument. To help uncover the biological mechanisms underlying this phenomenon, we used BPV1-specific PCR to test (i) whether BPV1 DNA is secreted by infected cells (ArcAi-1; P10) into culture medium, and whether this medium can be used for transfer of viral DNA to BPV1-free fibroblasts (ArcA). Intriguingly, a 1701 bp E2E5 region could be amplified from cell-free ArcAi-1 culture medium as well as from ArcA exposed to this medium and then passaged three times (Figure 6a). This finding was confirmed by BPV1 inverse PCR, revealing the presence of BPV1 episomes in cell-free ArcAi-1 medium as well as ArcA incubated therewith (Figure 6a,b).

ArcAi-1-medium-exposed ArcA were monitored over five additional passages. They resembled non-infected ArcA with respect to gross-morphology and growth characteristics, suggesting that the viral genomes could not be maintained at high levels. Of note, we failed to reproduce these data with cell-free culture medium from ArcAi-1 in P22. It is thinkable that BPV1 DNA secretion by infected cells decreases upon multiple passaging.

### 3.7. Extracellular Vesicles Secreted by Infected Cells Harbored Viral DNA

Given that secretion of BPV1 DNA by ArcAi-1 into culture medium could be shown, we next analyzed (i) whether this process would be (also) mediated by extracellular vesicles (EVs), and (ii) which EV fractions contain BPV1 DNA in this case. To this aim, cell-free ArcAi-1 culture media from P20 and P22 were subjected to differential ultracentrifugation, and DNA isolates from obtained fractions (remaining cell debris, fractions enriched for ABs, MVs, and EXOs) were PCR-screened for the presence of E5 (P22) or E6E7, i.e., a 1320 bp BPV1 region comprising the E6 and E7 open reading frames (ORFs; P20). E5 PCR scored positive for AB- and EXO-enriched fractions derived from ArcAi-1 in P22. E6E7 PCR yielded amplicons of expected size from all EV fractions, with Ethidium bromide staining of amplicons exhibiting a more pronounced signal for fractions enriched for ABs and EXOs (Figure 7a,b). Taken together, obtained PCR results indicated that BPV1 DNA predominated in apoptotic bodies (ABs) and exosomes (EXOs).

## 4. Discussion

Genetic analyses have mapped the transforming activities of BPV1 and BPV2 to the early genes E5, E6 and E7 [40]. These oncoproteins are expressed throughout sarcoid development and act in a combined manner. They assure the survival of infected host cells by inducing their immortalization, hyperproliferation and anoikis-independent growth [13,27,29]. Current in vitro models of sarcoid disease, i.e., primary sarcoid cells and BPV1-transfected equine fibroblasts authentically recapitulate these and other important features of BPV1 infection in horses and associated cell transformation [27,28,29,30]. However, both systems also have some limitations. Neither explanted sarcoid cells nor BPV1-transfected cells seem able to maintain BPV1 DNA upon freezing and thawing (S. Brandt; personal communication). In addition, both systems are unsuited to study early events in the viral lifecycle, notably the transit of virions from initial interactions with the cell surface to the delivery of viral DNA into the nucleus.

To overcome these limitations, we generated an in vitro model consisting of equine primary fibroblasts infected with wild-type BPV1 and BPV2 virions. Sarcoids are mainly diagnosed in young adult horses, and commonly affect the head, the ventral abdomen, and the paragenital region [41]. Thus, we established equine primary fibroblasts from skin of the eyelid, axilla, and prepuce of a 10-year-old sarcoid- and BPV1/-2 free gelding. Previous attempts to infect equine fibroblasts with BPV1 virions were unsuccessful [29,32,33]. Therefore, we focused on creating the best possible conditions for infection by (i) using fresh virion preparations, (ii) quantifying infectious virions, (iii) exposing primary fibroblasts to various virion concentrations, and (iv) exposing cells to virions in FBS-free culture medium. Successful infection was neither influenced by the virus type (BPV1 vs. BPV2) nor virion concentrations (20-2 Mio virions) or the origin of primary cells (eyelid, axilla, prepuce). We hence propose that use of fresh, confirmedly infectious virions and notably FBS-deprived medium were decisive factors of success. Heparan sulfate proteoglycans (HSPGs) are recognized attachment molecules on the cell surface required for initial PV attachment prior to infection [42,43,44]. Recently, it has been demonstrated that FBS can block HSPG-binding sites, thus impairing in vitro infection of HeLa or OVCAR4 cells by HPV16 pseudovirions [45].

BPV1- and BPV2-infected cells were monitored over up to 34 passages and several cycles of freezing and thawing with respect to growth characteristics, viral replication, and viral mRNA transcription. Infected cells consistently displayed sarcoid cell morphology, hyperproliferation and independence from anoikis, and harbored viral DNA and mRNA throughout experiments. None of these features was affected by multiple passaging or freezing and thawing. We hence propose that in vitro infection of primary equine fibroblasts with wild-type BPV1 and BPV2 virions yielded immortal cells closely resembling their natural counterparts.

Horses were for a long time regarded as dead-end hosts for BPV1 and BPV2, as infection was thought to be restricted to dermal fibroblasts that are not permissive for productive infection. As a consequence, bovines were thought to represent the only source of infectious virions [12]. With the advent of more sensitive molecular biological techniques, it could be shown that BPV1 infection also involves the equine epidermis [46,47] and can be productive in horses, albeit at a low level [15,48]. As herein reported, we could show that stable infection of equine fibroblasts can be achieved with only 20 infectious particles. Together with co-stabling experiments demonstrating direct transfer of infection from sarcoid-bearing to healthy donkeys [49], our findings further support the concept that BPV1 can spread within equid populations.

In our cell infection experiments, viral DNA loads initially correlated with virion concentrations (20 to 2 Mio virions) used for infection. From passage four onwards however, viral DNA copy numbers reached a constant level of about 150 copies/cell irrespective of virion concentrations to which cells were initially exposed. This finding points to the existence of a physiological viral replication level, at least in vitro. In equine sarcoid patients, BPV1 DNA loads were shown to range between <1 and 570 copies per tumor cell. Sarcoids of horses affected by mild disease harbored lower viral DNA copy numbers (<1.5 copies/cell), than lesions from severely affected horses (3–570 copies/cell) as determined by E2 qPCR [11]. We conclude that our infection model authentically recapitulates average viral replication levels seen in sarcoid patients.

The BPV1/2 genes E5 and L1 were transcribed in infected cells throughout passages. In horses, the major E5 oncoprotein is expressed throughout infection. It forms dimers and oligomers, and in this state, interacts with important host cell regulatory proteins, most notably the PDGFβ-R. Binding of E5 to this receptor leads to sustained activation of the latter—a prerequisite for E5-induced transformation of infected cells [50]. BPV1 E5 also inhibits cell surface expression of MHC class I, thus compromising viral antigen presentation and allowing BPV1 to escape from immune surveillance in the equid host [26]. On these grounds, consistent detection of E5 transcripts from in vitro infected cells was anticipated. However, we were surprised to detect L1 mRNA with the same consistency in infected fibroblasts. IC/PCR from cell extract and culture medium supernatants scored negative, thus refuting the possibility of virion production in these cells. We conclude that episomal BPV1/2 infection of equine fibroblasts allows for L1 transcription yet speculate that L1 protein expression might be inhibited by cellular factors, notably the inability of fibroblasts to differentiate in the way keratinocytes do.

The mechanisms underlying the spread of BPV1/infection within the horse’s integument are still unclear. There are several lines of evidence supporting that extracellular vesicles (EVs) are hijacked by viruses and contribute to virus propagation within an organism. For example, it has been shown that EVs from hepatitis C virus (HCV) -infected cells contain viral RNA promoting infection of normal cells [51]. The exosome pathway is also used by non-enveloped viruses for cell-to-cell spread and circumvention of the immune system [52]. Research on the role of EVs in PV infection and related tumour disease is still in its infancy. In 2021, there were less than 50 experimental papers available addressing the cargo and contribution of EVs to high-risk (hr) HPV-associated cancer development and progression [53]. The field of EVs in animal PV infection is completely unexplored. Herein, we show that BPV1-infected fibroblasts secreted viral DNA including full-length BPV1 genomes into culture medium. Furthermore, we provide evidence that this medium can be used for transfer of viral DNA to native fibroblasts. This led us to postulate that supernatant-mediated transfer of viral DNA from infected donor to native recipient cells might be achieved-at least in part-by EVs. Intriguingly, BPV1 PCR conducted from EV fractions secreted by infected cells revealed the presence of viral DNA in EVs, notably in fractions enriched for apoptotic bodies (ABs) and exosomes (EXOs). This finding agrees with previous reports on the presence of HPV nucleic acids and proteins in EXOs [53], and evidence that genetic information can be transferred by ABs from apoptotic to viable cells through the process of phagocytosis. This genetic material can include viral molecules, as shown for ABs released by Burkitt-derived lymphocytes: such ABs harbored Epstein–Barr virus (EBV) DNA and were able to transfer this DNA to professional and, importantly, non-professional phagocytes including fibroblasts [54]. On basis of these data and herein presented findings, we hypothesize that BPV1/2 infection is spread within the horse’s integument by EVs. Research addressing this intriguing possibility is ongoing.

## 5. Conclusions

Herein, we report on the successful and stable infection of equine primary fibroblasts with wild-type BPV1 and BPV2 virions. Infected cells displayed typical features of equine sarcoid cells, notably hyperproliferation, loss of contact inhibition, and probably immortalization. Infected cells contained viral episomes from which genes were consistently transcribed throughout passages and repeated cycles of freezing and thawing. Importantly, infected cells released BPV1 genomic DNA that was taken up by native fibroblasts. Intriguingly, we could also show that EVs secreted by BPV1-infected fibroblasts contained viral DNA, notably in the fractions enriched for ABs and EXOs.

We conclude that primary equine fibroblasts experimentally infected with wild-type BPV1 (or BPV2) virions show great potential in constituting an excellent near-natural model for the study of viral infection and spread, with particular emphasis on the role of EVs as potent vehicles for the transfer of viral molecules and other tumorigenic cargo from BPV1-infected to naive cells.

## Figures and Tables

**Figure 1 viruses-14-02658-f001:**
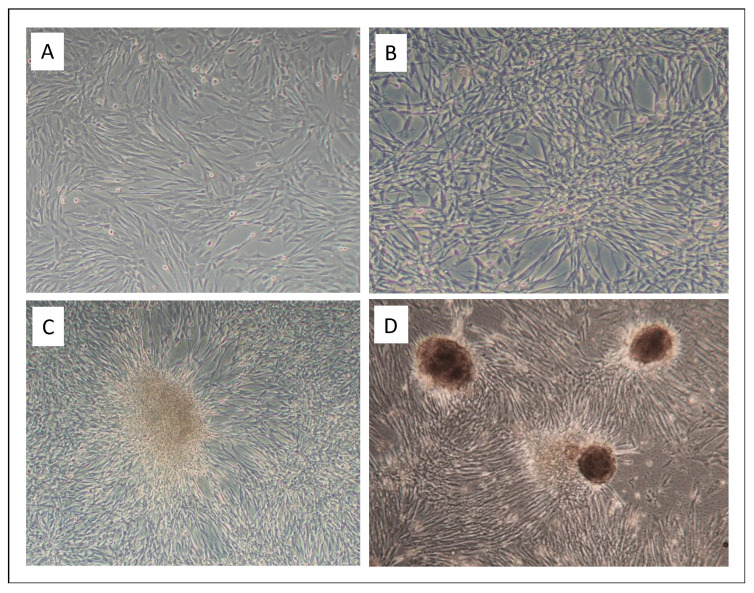
BPV1- and 2-infected fibroblasts lose contact inhibition and form aggregates. (**A**): non-infected equine primary fibroblasts (ArcA) in passage 3 (P3): the cells display the typical flat, elongated, spindle-shaped morphology of normal fibroblasts. (**B**): loss of contact inhibition in BPV1-infected equine fibroblasts in P3: cells begin to aggregate and pile up. C and D: BPV1- (**C**) and BPV2-infected equine fibroblasts (**D**) form dense three-dimensional cell aggregates consistent with tumor cell spheres. These events were documented by inverse phase-contrast microscopy (Axiovert 40 CFL, Zeiss, Jena, Germany) at 50× and 100× magnification.

**Figure 2 viruses-14-02658-f002:**
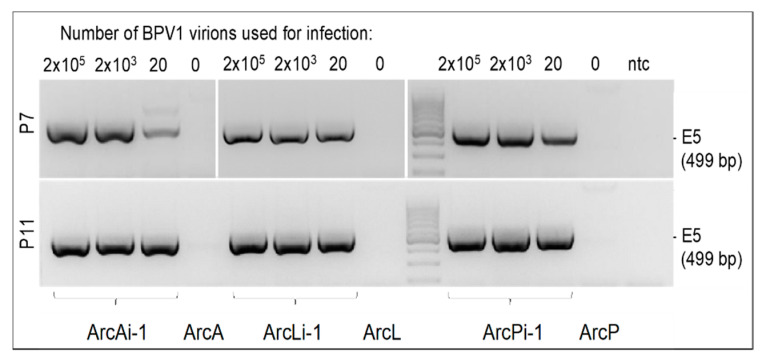
Infection of fibroblasts achieved with 20 to 2 × 10^5^ BPV1 virions. Standard consensus BPV1/2 E5 PCR from DNA of BPV1-infected versus non-infected equine fibroblasts from P7 and P11. L: 100 bp DNA Ladder (Thermo Fisher Scientific); ntc: no-template control (sterile water).

**Figure 3 viruses-14-02658-f003:**
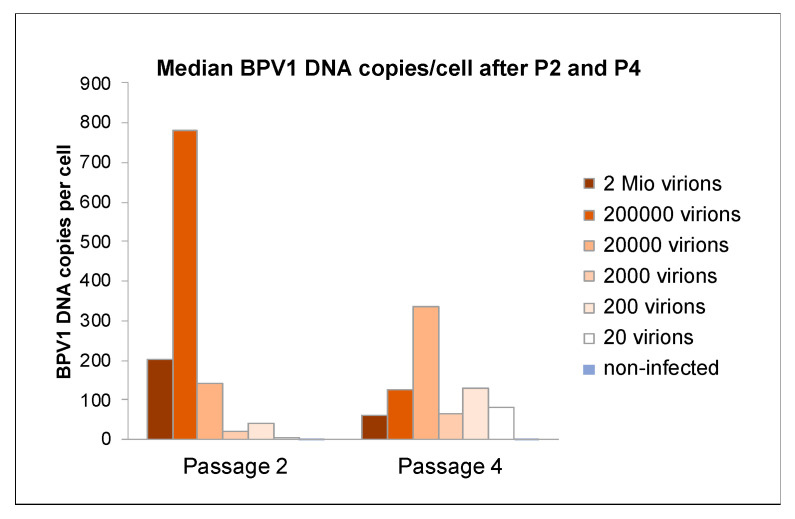
After four passages, BPV1 DNA loads settled at around 150 copies per cell. Medium BPV1 DNA loads were determined by E2 qPCR for ArcAi-1 at P2 and P4. The different colors of the bars represent respective amounts of virions used for initial infection as specified by the corresponding legend on the right side of the graph.

**Figure 4 viruses-14-02658-f004:**
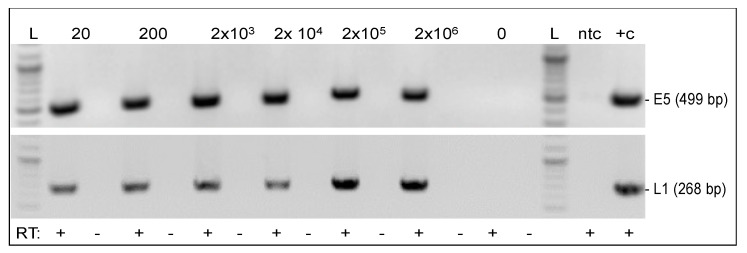
RT-PCR-mediated detection of BPV1 E5 and L1 mRNA from infected cells. BPV1 E5 and L1 RT-PCR from total, DNase-digested RNA of infected equine fibroblasts (P11). 20-2 × 10^6^/0: number of virions used for in vitro infection of cells; ntc: no-template control (sterile water); +c: positive control (sarcoid RNA); RT: +, −: reverse transcription performed with (+), or without (−) enzyme (controls). L: Gene Ruler DNA Ladder Mix (Thermo Fisher Scientific).

**Figure 5 viruses-14-02658-f005:**
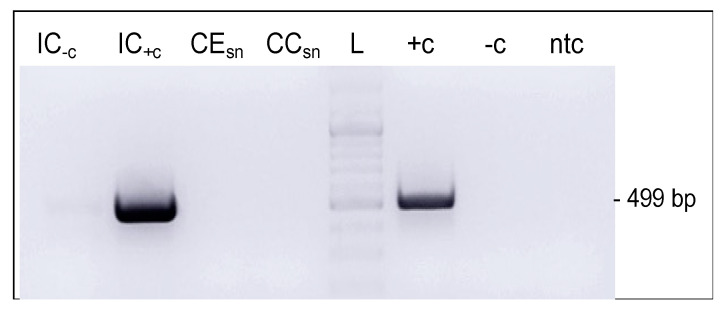
E5 IC/PCR fails to reveal virions in infected cells and culture medium. E5 IC/PCR using 5B6 as capture antibody. IC_−c_: 100 ng of BPV1 virion DNA in PBS as IC/PCR negative control; IC_+c:_ 2 × 10^4^ BPV1 virions in PBS as IC/PCR positive control; CE_sn_: cell extract supernatant from ArcAi-1 (P11); CC_sn_: cell culture supernatant from ArcAi-1 (P11); L: Gene Ruler DNA Ladder Mix (Thermo Fisher Scientific); +c: sarcoid DNA as PCR positive control; −c: normal horse skin DNA as PCR negative control; ntc: no-template control (sterile water).

**Figure 6 viruses-14-02658-f006:**
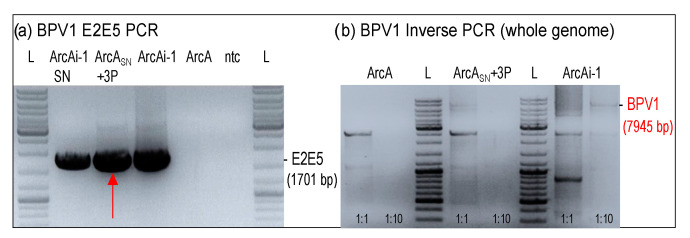
Transmission of BPV1 DNA by culture medium as revealed by BPV1 PCR. BPV1 E2E5 (**a**) and inverse PCR (**b**) from cell-free culture supernatant of ArcAi-1 (ArcAi-1 SN), ArcA cells exposed to this supernatant and then passaged for tree times (ArcA SN + 3P), BPV1 infected ArcA cells (ArcAi-1; positive control), native ArcA (ArcA; negative control), and sterile water (ntc). L: GeneRuler DNA Ladder Mix (Thermo Fisher Scientific).

**Figure 7 viruses-14-02658-f007:**
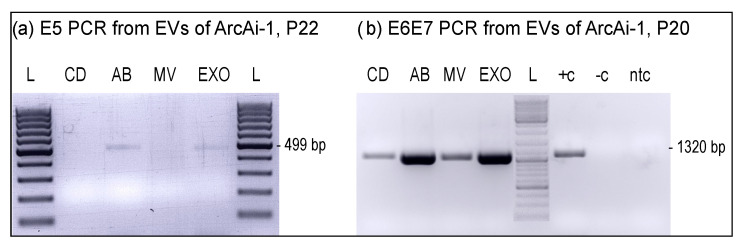
EVs, notably ABs and EXOs, harbor BPV1 DNA. BPV1 E5 (**a**) and E6E7 PCR from EV species isolated from cell-free culture supernatant of ArcAi-1 in P20 (**b**) and P22 (**a**). L: GeneRuler 100 bp DNA Ladder (Thermo Fisher Scientific); CD: cell debris; AB: apoptotic bodies; MV: microvesicles; EXO: exosomes; +c: positive control (sarcoid DNA); −c: negative control (normal equine skin DNA); ntc: no-template control (sterile water).

## Data Availability

All methods and major findings of the study are presented and discussed in the manuscript. Further information and raw data, notably with respect to growth characteristics of all infected cell preparations throughout passaging will be made available by the authors upon request.

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
