# Peer review of "Bovine Papillomavirus Type 1 or 2 Virion-Infected Primary Fibroblasts Constitute a Near-Natural Equine Sarcoid Model"

_viruses, 2022, doi:10.3390/v14122658_

Round 1

Reviewer 1 Report (Previous Reviewer 2)

The main problem of this manuscript is that the authors claim something about the properties of virus-infected cells, but they have not performed any standard methods that justify the claims in the paper.

Author Response

We conducted the presented research according to standard procedures on the basis of our experience in BPV research and did our best to implement suggestions of Reviewer 1.

Reviewer 2 Report (Previous Reviewer 1)

The reviewer's concerns have been addressed

Author Response

We warmly thank this reviewer for approval of our manuscript.

Reviewer 3 Report (New Reviewer)

This is a very well designed study and the results are clearly presented and appropriately discussed.

Comments:
1. It would be interesting to discuss further why innoculation of the fibroblast cultures with BPV-1 and BPV-2 was successful in this study, where it has failed in all previous attempts. Indeed, this has held up the study of sarcoid pathogenesis for many years. It would be interesting to discuss what might be happening in 'fresh' cow papilloma samples compared to those that are older and why fetal calf serum might be an inhibitor of infection.
2. It is not clear why only BPV-1 and not BPV-2 was used in cell-free infection experiments. The discussion clearly infers that BPV-2 infection is equivalent to BPV-1 infection, and that the modes of transmission between cells and individuals is the same. However, the cell-free transmission and extravesicular experiments only refer BPV-1 and not BPV-2.

Author Response

Ad 1: We thank this reviewer for this interesting comment. It motivated us to conduct further literature search, leading to the discovery of a very recent article describing inhibition of PV infection by FBS via blockade of virion attachment molecules!  The corresponding passage in the discusion has been expanded accordingly, including appropriate citations. It now reads:

".......We hence propose that use of confirmedly infectious virions and notably FBS-deprived medium during infection were decisive factors of success. Heparan sulfate proteoglycans (HSPGs) are recognized attachment molecules on the cell surface required for initial PV virion attachment prior to infection [42-44]. Recently, it has been demonstrated that FBS can block HSPG-binding sites, thus impairing in vitro infection of HeLa or OVCAR4 cells by HPV16 pseudovirions [45].  (see also Lines 383-388 in the revised manuscript).

With respect to freshness of virions: We would like to stress that use of confirmedly infectious virions contributed to success (with fresh preparations probably containing a higher percentage of non-degraded infectious particles). 

To avoid misunderstandings, we have deleted the term "fresh". (Line 383)

Ad 2: We fully agree. We did not include BPV2 in these experiments due to considerable financial restrictions. 

Thank you very much for the positive reception of our paper.

This manuscript is a resubmission of an earlier submission. The following is a list of the peer review reports and author responses from that submission.

Round 1

Reviewer 1 Report

The paper aimed to describe the successful infection of equine primary fibroblasts with wild-type BPV1 and BPV2 virions, which falls within the general scope of Viruses. This is an interesting paper, which focuses on the study of infected fibroblasts with respect growth characteristics, stability of infection, viral gene transcription, viral DNA secretion, and release of extracellular vesicles harboring viral DNA. The paper is well-written, the methodology is sound and the Results are well discussed.

Some small concerns:

There are some comments from the authors in the file that must be removed.

Lines 45-46: This statement is confusing! The authors mistakenly use the term “homology” instead of “similarity” or “identity”. Homology is a concept from evolutionary biology, which means that two sequences/characters have a common evolutionary origin (Reeck et al., 1987). It is a qualitative concept. In addition, the authors state “<90% on L1 capsid protein level”, which does not seem accurate!

Reviewer 2 Report

Sarcoids are the most common tumors in equids; their association with bovine papillomaviruses (BPV) infection has been widely reported, but the mechanism of carcinogenesis has not been fully elucidated. This manuscript therefore presents a n in vitro model based on BPV virion infection of equine fibroblasts. The authors successfully infected fibroblasts and detected viral DNA and RNA during several passages of equine fibroblasts. However, I do not think that the experiments presented are sufficient to reach the conclusions that the authors present

I have few minor comments:

in the context of monitoring the presence of BPV DNA in EVs, it would certainly be advantageous to add in the chapter “Screening for transmission of infection” how the medium from confirmedly BPV1-infected ArcA cells were filtered to remove cells and cell debris.

In the text are comments to the authors, will some of them be addressed?

How can the authors explain the results given in the legend to the figure 2 in line 260-261 “The slight band displayed by non-infected ArcL from P7 was due toa pipetting error. Repetition of the experiment for ArcLi-1 and ArcL (P7) DNA produced a negative result for ArcL (not shown)“

Primers used for detection of viral transcripts are used for DNA detection in the original article. How did the authors treat the possibility of viral RNA contaminating by its DNA genome?

How were the fractions after ultracentrifugation characterized? The authors mentioned remaining cell debris, fractions enriched for ABs, MVs, and EXOs? The difference between the extracellular vesicles is quite small and it is not easy to distinguish between them.

Legend to the Figure one is not sufficient.

And regarding the topic of the manuscript, I have the major comments:

In chapter 3.1. “Infection of equine primary fibroblasts with BPV1 and -2 virions induced hyperproliferation, loss of contact inhibition and - most probably – immortalization”, the author mentioned the hyperproliferation, loss of contact inhibition and cell aggregation, however they showed only the picture of BPV infected cells not the control non-infected cells.

Here I miss some sophisticated methods (for example MTT assay, wound healing assay, clonogenic assay) that accurately determine the proliferation and migration of cells. I do not consider the images from infected cells only to be sufficient to support the conclusions the authors draw regarding infected cells. Yet when the authors suggest the possibility of immortalization.

In this context, I also don't understand the last sentence in lines 241-243. Could the authors explain this sentence?

Could the authors explain the sentence in line 265-266 “At passage 2 (P2), virion concentrations used for infection directly correlated with cell proliferation rates: The higher the virion concentration used for infection, the higher were BPV1 E2 copy numbers per cell, as determined by qPCR. How the BPV E2 copy correlated with cell proliferation? How was the cell proliferation rate estimated?

How it is with the BPV1 DNA load in infected cells after P4 especially when you reach P34? What about the stability of the viral genome in infected cells during later passages?